# Lymphocyte predominant cells detect *Moraxella catarrhalis*-derived antigens in nodular lymphocyte-predominant Hodgkin lymphoma

Lorenz Thurner [1,16✉], Sylvia Hartmann[2,16], Natalie Fadle[1], Evi Regitz[1], Maria Kemele[1], Yoo-Jin Kim[3], Rainer Maria Bohle[3], Anna Nimmesgern[4], Lutz von Müller[5], Volkhard A. J. Kempf[6], Marc A. Weniger[7,8], Frank Neumann[1], Nadine Schneider[9], Martine Vornanen[10], Christer Sundström[11], Laurence de Leval[12], Andreas Engert[13,14], Dennis A. Eichenauer[13,14], Ralf Küppers [7,8], Klaus-Dieter Preuss[1], Martin-Leo Hansmann[2,15,17] & Michael Pfreundschuh[1,17]

Nodular lymphocyte-predominant Hodgkin lymphoma (NLPHL) is a rare lymphoma of B-cell origin with frequent expression of functional B-cell receptors (BCRs). Here we report that expression cloning followed by antigen screening identifies DNA-directed RNA polymerase beta' (RpoC) from *Moraxella catarrhalis* as frequent antigen of BCRs of IgD[+] LP cells. Patients show predominance of HLA-DRB1*04/07 and the IgVH genes encode extraordinarily long CDR3s. High-titer, light-chain-restricted anti-RpoC IgG1/κ-type serum-antibodies are additionally found in these patients. RpoC and MID/hag, a superantigen co-expressed by *Moraxella catarrhalis* that is known to activate IgD[+] B cells by binding to the Fc domain of IgD, have additive activation effects on the BCR, the NF-κB pathway and the proliferation of IgD[+] DEV cells expressing RpoC-specific BCRs. This suggests an additive antigenic and superantigenic stimulation of B cells with RpoC-specific IgD[+] BCRs under conditions of a permissive MHC-II haplotype as a model of NLPHL lymphomagenesis, implying future treatment strategies.

[1] José Carreras Center for Immuno- and Gene Therapy and Internal Medicine I, Saarland University Medical School, Homburg/Saar, Germany. [2] Dr. Senckenberg Institute of Pathology, Goethe University, Frankfurt am Main, Germany. [3] Saarland University Medical School, Institute of Pathology, Homburg/Saar, Germany. [4] Institute of Medical Microbiology and Hygiene, University of Saarland, Homburg, Germany. [5] German National Reference Center for Clostridoides (Clostridium) difficile, Institute for Laboratory Medicine, Microbiology and Hygiene, Christophorus Kliniken, Coesfeld, Germany. [6] Institute of Medical Microbiology and Infection Control, University Hospital of the Goethe University, Frankfurt am Main, Germany. [7] Institute of Cell Biology (Cancer Research), University of Duisburg-Essen, Medical School, Essen, Germany. [8] Deutsches Konsortium für translationale Krebsforschung (DKTK), Heidelberg, Germany. [9] Fraunhofer Institute for Molecular Biology and Applied Ecology IME, Project Group Translational Medicine & Pharmacology, and Division of Rheumatology, University Hospital Frankfurt Goethe University, Frankfurt am Main, Germany. [10] Department of Pathology, Tampere University Hospital and University of Tampere, 33520 Tampere, Finland. [11] Department of Immunology, Genetics and Pathology, Uppsala University Hospital, Uppsala, Sweden. [12] Institute of Pathology, Lausanne University Hospital, Lausanne, Switzerland. [13] University of Cologne, First Department of Internal Medicine, Center for Integrated Oncology Aachen Bonn Cologne Dusseldorf, Cologne, Germany. [14] German Hodgkin Study Group, First Department of Internal Medicine University Hospital Cologne, Cologne, Germany. [15] Frankfurt Institute of Advanced Studies, Frankfurt am Main, Frankfurt, Germany. [16]These authors contributed equally: Lorenz Thurner, Sylvia Hartmann. [17]These authors jointly supervised this work: Martin-Leo Hansmann, Michael Pfreundschuh. ✉email: lorenz.thurner@uks.eu

odular lymphocyte-predominant Hodgkin lymphoma (NLPHL) accounts for 5–16% of all Hodgkin lymphoma (HL) cases[1], and IgD[+] NLPHL represents a distinct clinical subtype, with a strong (>20:1) male predominance[2]. The disease-defining lymphocyte predominant (LP) tumor cells represent only a small proportion of the total tumor infiltrate and are widely outnumbered by reactive cells. LP cells have a late germinal-center B-cell phenotype and are closely related to the tumor cells of T-cell/histiocyte-rich large B-cell lymphoma (THRLBCL)[3,4]. LP cells harbor recurrent mutations in *SGK1*, *DUSP2*, *JUNB*, and *SOCS1*, frequent translocations affecting the B-cell lymphoma 6 (*BCL6*) gene, usually have a preserved B-cell phenotype, and often express a functional B-cell receptor (BCR)[3,5–10]. Intraclonal immunoglobulin variable (IgV) region gene diversification[7], the strong expression of BCL6[11], and activation-induced cytidine deaminase (AID)[12,13], and a histological picture that resembles germinal centers indicate an ongoing immune response. Activation of the nuclear factor (NF)-κB pathway is observed in LP cells[3], but NF-κB activity in LP cells is neither caused by mutations in *NFKBIA* or *TNFAIP3*[14] nor by Epstein-Barr virus (EBV)[15] infection and, thus, could represent a consequence of chronic BCR stimulation. Because IgD[+] NLPHL has a predilection for cervical lymph nodes, we addressed the question of whether bacteria in the upper respiratory tract play a role in the pathogenesis of this lymphoma. To identify the antigenic targets of BCRs expressed on LP cells, recombinant antigen-binding fragments (Fabs) were constructed and screened for their ability to bind with pathogens and auto-antigens.

Here we report that IgD[+] LP cell-derived Fabs bind the *Moraxella catarrhalis* antigen DNA-directed RNA polymerase beta' (RpoC) and thus reveal together with a permissive HLA class II haplotype of the patients and presence of light-chain-restricted serum antibodies a bacterial infection as trigger for the lymphomagenesis of IgD[+] NLPHL.

## Results

**Patients and Ig V gene characteristics**. Functional Ig heavy and light chain genes were successfully amplified from microdissected LP cells from 12 of 22 NLPHL cases of a screening cohort from Germany and Finland, including two composite lymphomas consisting of an NLPHL part (case #7a and #8a) and a diffuse large B cell lymphoma (DLBCL) part (case #7b and #8b) in the same lymph node. In a validation cohort, composed of only IgD[+] NLPHL cases from Switzerland and Sweden, the success rate was 3/5 (#13-#15). The median age of patients with successfully amplified IgV genes was 30 years. Eight patients had IgD[+] LP cells (Supplementary Fig. 1), and five of these patients were adolescents (Table 1). Two IgD[+] NLPHL samples were obtained from inguinal lymph nodes, but these were relapses. A male predominance was observed among the NLPHL cases (13 of 15). All cases had mutated IgV genes, with mutation frequencies for heavy and light chain IgV genes ($V_H$ and $V_L$, respectively) ranging between 0% and 18.0% (average: 8.3% for $V_H$ and 4.8% for $V_L$ gene segments; Table 2). The complementarity determining region (CDR) 3 of $V_H$ region genes isolated from IgD[+] LP cells were significantly longer (median: 30 amino acids; mean: 29.95 ± 1.048 [SEM] amino acids; $n = 8$), compared with the CDR3 of $V_H$ region genes isolated from IgD[−] LP cells (median: 17 amino acids; mean: 17.71 ± 1.017 amino acids; $n = 7$; $p < 0.0001$, unpaired two-tailed Student's *t*-test). Seven of the eight IgD[+] NLPHL cases expressed a member of the $V_H$3 family, compared with one of the seven IgD[−] NLPHL cases. Furthermore, cases with extraordinarily long CDR3 had characteristic VDJ-recombinations (frequently D3-3*01-JH6 rearrangements, see Table 2).

**BCRs derived from IgD[+] LP cells react with *Moraxella* spp**. Overall, 6/8 IgD[+] NLPHL-derived BCRs (four [#3, #6, #9, and #10] from the screening cohort and two [#13 and #14] from the validation cohort) reacted against *M. catarrhalis* lysates (Fig. 1a)

**Table 1 Characteristics of the NLPHL patients included in this study.**

| Case | Gender | Age (year) | Disease status | Localization | LP cells IgD[+] | Reactivity against *M. cat.* RpoC | Reactivity against *M. osl.* Succinate-CoA ligase sub α | HLA-DRB1*04 or HLA-DRB1*07 |
|---|---|---|---|---|---|---|---|---|
| 1 | m | 52 | Relapse | Axillary | No | No | No | No |
| 2 | m | 15 | Relapse | Supraclavicular | No | No | No | No |
| **3** | **m** | **14** | **2nd relapse** | **Cervical** | **Yes** | **yes** | **No** | **Yes** |
| 4 | m | 65 | First manifestation | Cervical | No | No | No | Yes |
| 5 | f | 37 | First manifestation | Retroperitoneal | No | No | No | No |
| **6** | **m** | **15** | **Relapse** | **Cervical** | **Yes** | **Yes** | **No** | **Yes** |
| 7a | m | 51 | 2nd relapse | Axillary | No | No | No | No |
| 7b | m | 51 | 2nd relapse | Axillary | No | No | No | No |
| 8a | m | 40 | Primary progressive | Abdominal | No | No | No | No |
| 8b | m | 40 | Primary progressive | Abdominal | No | No | No | No |
| **9** | **m** | **42** | **Relapse** | **Inguinal** | **Yes** | **Yes** | **No** | **Yes** |
| **10** | **m** | **12** | **2nd relapse** | **Inguinal** | **Yes** | **Yes** | **No** | **Yes** |
| **11** | **m** | **15** | **First manifestation** | **Cervical** | **Yes** | **No** | **Yes** | **Yes** |
| 12 | m | 31 | First manifestation | Cervical | No | No | No | Yes |
| **13** | **m** | **18** | **First manifestation** | **Cervical** | **Yes** | **Yes** | **No** | **Yes** |
| **14** | **m** | **30** | **First manifestation** | **Parotid** | **Yes** | **Yes** | **No** | **Yes** |
| **15** | **f** | **16** | **First manifestation** | **Axillary** | **Yes** | **No** | **No** | **No** |

Bold: IgD[+] NLPHL; cases 7 and 8: (a) NLPHL, (b) histological transformation into DLBCL in the same lymph node.

**Table 2 Ig variable region gene analysis of LP cells.**

| Case | LP cells IgD+ | VH/VL gene | Homology (%) | Somatic mutation | JH/JL gene | DH gene | length CDR3 (AA) | Junction |
|---|---|---|---|---|---|---|---|---|
| 1 | Neg | VH1-3-01 | 91.3 | Yes | JH4*02 | D3-10*01 | 20 | CAREVRPPRIIMIWGVGLLDFW |
| 1 | Neg | VK1-27*01 | 97.5 | Yes | JK4*01 | – | 9 | CQKYNSAPLTF |
| 2 | Neg | VH1-46*01 | 87.5a | Yes | JH2*01 | D3-16*02 | 15 | YYCARDEGDIRRYFDLW |
| 2 | Neg | VK3-15*01 | 92.8 | Yes | JH5*01 | – D3-3*01 | 10 | CQQYNYWPPVTF |
| **3** | **Pos** | **VH3-11*01** | **95.1** | **Yes** | **JH6*03** | **D3-3*01** | **26** | **CARVAGAAGRNYNYWSGYWEDYYFMDVW** |
| **3** | **Pos** | **VK2D30*01** | **99.3** | **Yes** | **JH4*01** | **-** | **9** | **CMQGTHWPR** |
| 4 | Neg | VH4-31*01 | 99.2 | Yes | JH3*02 | D3-22 | 19 | CARGPPPYDSSGYYSHGLDIW |
| 4 | Neg | VK1-37 | 99.3 | Yes | JK3*01 | – | 8 | GQRTYNAPRF |
| 5 | Neg | VH4-39*01 | 100 | No | JH6*04 | D6-19*01 | 15 | CASMGAVAGMMFGMDVW |
| 5 | Neg | VK1-5*03 | 96 | Yes | JK1*01 | – | 8 | CQEYNSYWTF |
| **6** | **Pos** | **VH3-07*01** | **95.5** | **Yes** | **JH6*04** | **D3-3*01** | **26** | **CAREVLRWGGSYDFWSNYYEDYFALDVW** |
| **6** | **Pos** | **VK-1D43*01** | **98.5** | **Yes** | **JK3*01** | **-** | **10** | **CQQYYSTPPFTF** |
| 7a | Neg | VH1-69 | 86.4 | Yes | JH5*02 | D6-19*01 | 17 | CARDYSRGVCGPRYGMDVW |
| 7a | Neg | VK3-11*01 | 93.8 | Yes | JK4*01 | – | 9 | CQQRSNWPPAF |
| 7b | Neg | VH1-69 | 84.7 | Yes | JH5*02 | D6-19*01 | 17 | CAREYSRGVCGPRYGMDVW |
| 7b | Neg | VK3-11*01 | 90.3 | Yes | JK4*01 | – | 9 | CQQRSNWPPAF |
| 8a | Neg | VH3-30 | 89.9 | Yes | JH6*02 | D2-15*01 | 22 | CARKGGDPVLALFVPNFAMDVW |
| 8a | Neg | VK1-33*01 | 91.6 | Yes | JK4*01 | – | 9 | CQQYNSLPITF |
| 8b | Neg | VH3-30 | 89.0 | Yes | JH6*02 | D2-15*01 | 22 | CARKGGDPVLAVFVPNFALNVW |
| 8b | Neg | VK1-33*01 | 91.1 | Yes | JK4*01 | – | 9 | CQQYNSLPITF |
| **9** | **Pos** | **VH3-48*03** | **82.0** | **Yes** | **JH6*02** | **D3-3*01** | **30** | **CAKSVLTAKSGKSYKFWNNYHEDYHYYLMDVW** |
| **9** | **Pos** | **VK1-27*01** | **87.1** | **Yes** | **JK3*01** | **-** | **10** | **CQNYNTVPLTF** |
| **10** | **Pos** | **VH4-59*01** | **86.4b** | **Yes** | **JH6*03** | **D3-3*01** | **33** | **CATVDPTVVEGRVKYYDFWSGYYGTDQRYYYMDVW** |
| **10** | **Pos** | **VL3-21*01** | **92.8** | **Yes** | **JL2*01** | **-** | **11** | **CQVWDSSSDHPVF** |
| **11** | **Pos** | **VH3-11*01** | **95.5** | **Yes** | **JH6*02** | **D3-3*01** | **30** | **CARLLTSEGSRKYYDFWSNYWEGYQYYTMDVW** |
| **11** | **Pos** | **VK2-28** | **99.2** | **Yes** | **JK3*03** | **-** | **9** | **CMQGLQTVFTF** |
| 12 | Neg | VH4-34 | 100 | No | JH5*02 | D3-10*01 | 16 | CARGPYLWFGERGWFDPW |
| 12 | Neg | VK1-5*01 | 94 | Yes | JK2*03 | – | 9 | CQQYNSYPYCF |
| **13** | **Pos** | **VH3-11*01** | **83.0** | **Yes** | **JH6*03** | **D3-3*01** | **26** | **CARLCAAGGRSYDFWSGYYENYFYMEVW** |
| **13** | **Pos** | **VK1-5*01** | **86.7** | **Yes** | **JK4*01** | **-** | **10** | **CQQYSGSSRVTF** |
| **14** | **Pos** | **VH3-11*01** | **87.6** | **Yes** | **JH6*02** | **D3-3*01** | **30** | **CARLIEAGGVGKHYDFWSGYYTVDYYYGMDVW** |
| **14** | **Pos** | **VK1-43*01** | **94.3** | **Yes** | **JK4*01** | **-** | **10** | **CQQYYSTPPLTF** |
| **15** | **Pos** | **VH3-30*04** | **89.8** | **Yes** | **JH6*02** | **D3-3*01** | **33** | **CARTTWVGVVGRIKYYDFWSGYHGTGMEYYTMDVW** |
| **15** | **Pos** | **VL9-49*01** | **98.1** | **Yes** | **JL3*02** | **–** | **12** | **CGADHGSGSNFWVF** |
| **15** | **Pos** | **VK1-17*01** | **94.33** | **Yes** | **JK4*01** | **–** | **10** | **CLQHNSYPRLTF** |

Bold: IgD+ NLPHL; all cases had functional IgV genes. For case 15 two functional light chains were amplified. For somatic mutation a threshold of 0.5% was used.
a Insertion in FR3 (AGAAAT).
bWith deletion of 3 nt in CDR2.

and the *M. catarrhalis* strain ATCC 43617 RO 108 (data not shown), whereas one IgD+ NLPHL (#11) reacted with a *M. osloensis* lysate (Fig. 1a, Supplementary Information and Supplementary Fig. 2). No BCRs derived from IgD− NLPHL cases reacted with either *M. catarrhalis* or *M. osloensis*. In Western blots of *M. catarrhalis* lysates, the BCRs derived from IgD+ NLPHL cases detected a target antigen of 150–160 kDa (Fig. 1b, Supplementary Figs. 3 and 4). *M. catarrhalis* outer membrane vesicles contain two potential candidate antigens that have molecular weights >150 kDa: *Moraxella* IgD-binding protein (MID/hag) and DNA-directed RNA polymerase subunit beta' of *M. catarrhalis* (RpoC)[16]. The MID/hag monomer has a molecular weight of 200 kDa, whereas the multimer has a molecular weight of ~800 kDa under non-reducing conditions (Supplementary Fig. 3)[17,18]. In contrast, the *Moraxella*-reactive BCRs (#3, #6, #9, and #10) recognized a protein of 150–160 kDa (Supplementary Fig. 4), which excludes MID/hag as their target. Moreover, the *Moraxella*-reactive BCRs did not bind to recombinant fragments of MID/hag in Western blots (data not shown). The specific binding of the BCRs with recombinant RpoC was demonstrated, both by ELISA and Western blots (Fig. 1b, c), confirming that RpoC was the 150–160 kDa band detected in the *M. catarrhalis* lysate. The dominant epitope of RpoC-reactive BCRs was spanning from amino acid 600–1130 of RpoC, which could be further narrowed down to amino acids 851–865 (Fig. 1d and Supplementary Fig. 5).

IgD+ status of NLPHL was significantly associated with reactivity of recombinant Fabs against *M. species* ($p = 0.007$;

Fisher's exact test) and *M. catarrhalis* RpoC ($p = 0.0014$; Fisher's exact test). In contrast to NLPHL, none of the recombinant BCRs derived from 12 DLBCL, 11 primary central nervous system lymphomas, 10 mantle cell lymphomas, and 12 chronic lymphocytic leukemia cases reacted with *M. catarrhalis* RpoC (Supplementary Fig. 6). Additionally, for IgD− NLPHL two autoantigens were identified as targets of BCRs of LP-cells (Supplementary Fig. 7). Several representative dot and western blots are shown in supplementary (Supplementary Figs. 16–22).

**Restriction to certain MHC II classes in IgD+ NLPHL.** Seven of the eight IgD+ NLPHL patients, including all six cases with an RpoC-specific BCR, had HLA-DRB1*04 or HLA-DRB1*07 haplotypes (Table 3). The presence of HLA-DRB1*04 or HLA-DRB1*07 was significantly associated with reactivity against *Moraxella* species ($p = 0.0256$; two-tailed Fisher's exact test), and with reactivity against *M. catarrhalis* RpoC ($p = 0.044$; two-tailed Fisher's exact test). For *M. catarrhalis* RpoC five T-cell epitopes, with high SYFPEITHI scores, were predicted for HLA-DRB1*04, and six epitopes were predicted for HLA-DRB1*07 (Supplementary Table 2), suggesting that these haplotypes can provide cognate follicular T-cell help for RpoC-specific B cells.

**NLPHL patients show high-titer anti-RpoC-serum antibodies.** Serum antibodies against *M. catarrhalis* RpoC were detected in 2/2 patients with IgD+ RpoC-specific BCRs within the screening cohort, at a titer of 1:3200, which belonged to the IgG1/κ subclass

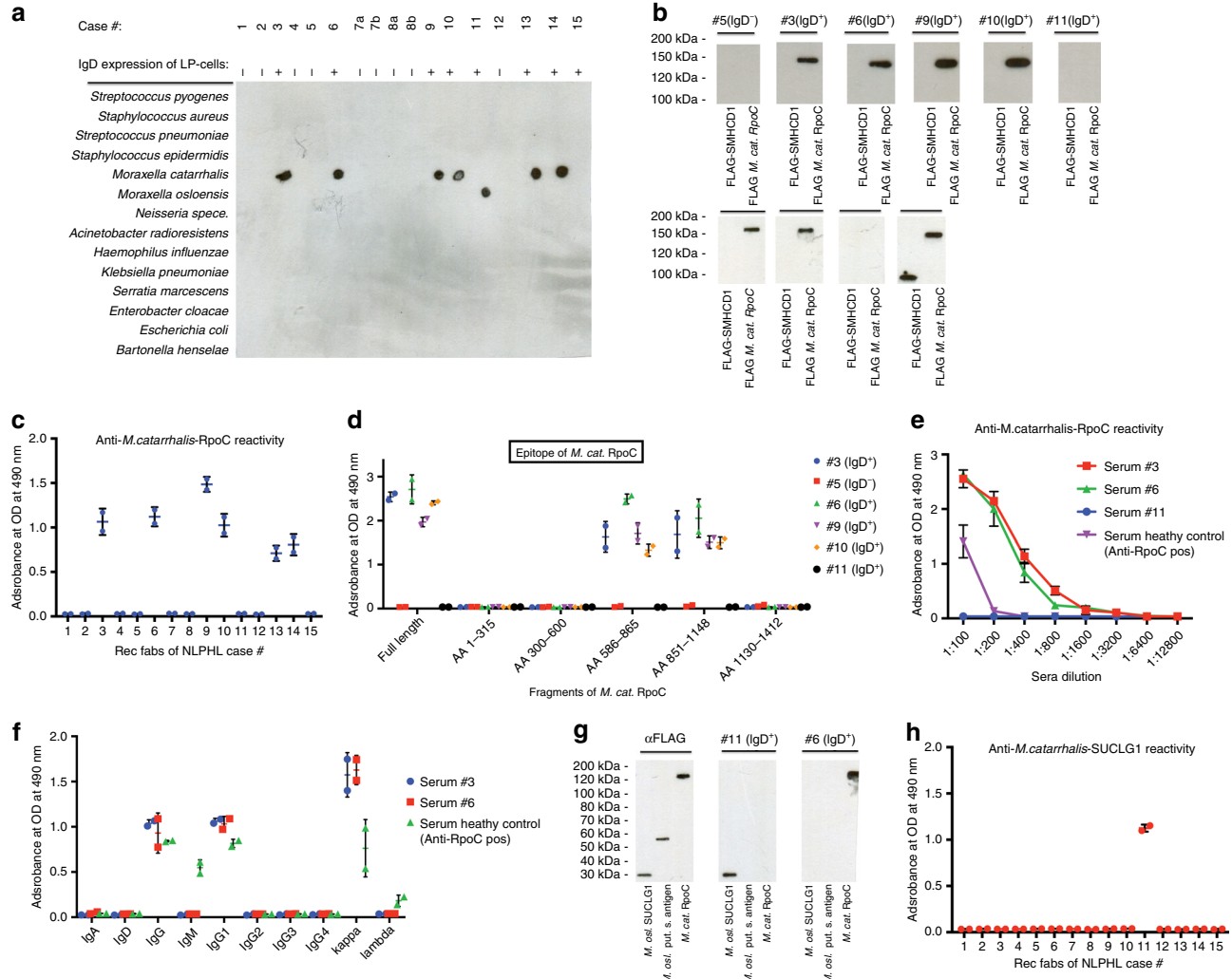

**Fig. 1 Reactivity of recombinant IgD+ LP cell-derived Fabs with bacterial lysates. a** Representative Immuno-dot blots of bacterial lysates. NLPHL-derived recombinant Fabs of six cases bound to lysates of *M. catarrhalis*, and of one case to lysate of *M. osloensis*. Eight cases showed no reactivity against bacterial lysates. All six *M. catarrhalis*-reactive recombinant Fabs (#3, #6, #9, #10, #13, and #14) and the one *M. osloensis*-reactive Fab (#11) were derived from IgD+ NLPHL cases. **b** Representative Western blot of recombinant C-terminally FLAG tagged *M. catarrhalis* RpoC expressed in HEK293 cells confirmed reactivity of lymphoma Fabs #3, #6, #9, #10, #13, and #14 with the recombinant bacterial antigen. A recombinant C-terminally FLAG-tagged SMCHD1 fragment was used as negative control. **c** ELISA with recombinant C-terminally FLAG-tagged *M. catarrhalis* RpoC as a coat demonstrated specific reactivity of Fabs derived from LP cells of six patients (#3, #6, #9, #10, #13, and #14) against recombinant RpoC. Values are mean ± SD. **d** Determination of the epitope of RpoC recognized by IgD+ NLPHL Fabs. Recombinant Fabs #3, #6, #9, and #10 bind AA 600–1130 of *M. catarrhalis* RpoC. Values are mean ± SEM. **e** Serum reactivity against RpoC in patients from whom recombinant NLPHL Fabs were derived. The two IgD+ NLPHL patients with RpoC-reactive recombinant lymphoma Fabs (#3 and #6) had higher anti-RpoC-antibody serum titers (1:1600) than a healthy control with anti-RpoC-antibodies (1:200). The IgD+ NLPHL patient with the *M. osloensis*-specific Fab (#11) had no serum anti-RpoC-antibodies. Values are mean ± SD. Experiments were independently repeated three times. **f**: Immunoglobulin classes and subclasses and light chains of serum RpoC antibodies from the two patients with RpoC-reactive BCR of the LP cells (#3 and #6) and a seropositive healthy control. Patients #3 and #6 both show a predominance of IgG1/κ. The healthy control had additionally anti-RpoC-antibodies of IgM class and in contrast to patients #3 and #6 no clear light chain restriction. Values are mean ± SD.
**g** Representative Western blot of FLAG-tagged *M. osloensis* SUCLG1 expressed in HEK293 cells confirms reactivity of lymphoma Fab #11 with the recombinant bacterial antigen. **h** ELSIA with recombinant *M. osloensis* SUCLG1 as a coat demonstrated specific reactivity of Fabs derived from LP cells of patient #11 against recombinant SUCLG1. Values are mean ± SD. Data in **a**–**c**, **e**, and **f** are representative of three independent experiments each; data in **d**, **g**, and **h** are representative of two independent experiments.

(Fig. 1e). High-titer, anti-*M. catarrhalis*-RpoC antibodies were found in 20/98 NLPHL patients enrolled in clinical trials conducted by the German Hodgkin Study Group (GHSG) (Fig. 2a), which is representative of the general population of NLPHL patients. All anti-*M. catarrhalis*-RpoC antibodies were of IgG class and predominantly of the IgG1 subclass (Fig. 2b), thus probably not related to the IgD+ LP cell clone. However, in each patient with serum antibodies against RpoC, these were light chain restricted (Fig. 2b), indicating a clonal origin and presumably a

relation to the initiating event of NLPHL. The titers ranged from 1:800 to 1:3,200 (Fig. 2c). When applying a titer higher than 1:400 as cut off for positivity, no healthy control or patient with cHL or THRLBCL was positive. High titer (>1:400) serum antibodies against *M. catarrhalis* RpoC were associated with NLPHL ($p <$ 0.0001; Fisher's exact test). Low-titer RpoC-antibodies were detected in the serum of 9/188 healthy controls (Supplementary Fig. 8) and in 10/100 patients with classical HL (Fig. 2d), with maximum titers of 1:200 (Fig. 2e, Supplementary Figs. 9 and 10).

**Table 3 MHC alleles of NLPHL.**

| Case | HLA-A | HLA-B | HLA-DRB1 | HLA-DQB1 | HLA-DQA1 |
|---|---|---|---|---|---|
| 1 | A*11/A*24 | B*44/B*51 | DRB1*01:01:01/DRB1*11:01:01 | DQB1*03:01:01/DQB1*05:01:01 | DQA1*01:05:01/DQA1*05:05:01 |
| 2 | A*01/A*02 | B*40/B*49 | DRB1*01:01:01/DRB1*11:02:01 | DQB1*03:19/DQB1*05:01:01 | DQA1*01:05:01/DQA1*05:05:01 |
| **3** | **A*03/A*32** | **B*44/B*51** | **DRB1*04:01:01/DRB1*11:01:01** | **DQB1*03:01:01/DQB1*03:02:01** | **DQA1*03:01:01/DQA1*05:05:01** |
| 4 | A*02/A*31 | B*18/B*56 | DRB1*04:01:01/DRB1*15:01:01 | DQB1*06:02:01/DQB1*03:02:01 | DQA1*01:02:01/DQA1*03:01:01 |
| 5 | A*03/A*32 | B*15/B*47 | DRB1*11:01:01/DRB1*11:01:01 | DQB1*03:01:01/DQB1*03:02:01 | DQA1*03:01:01/DQA1*05:05:01 |
| **6** | **A*30/A*32** | **B*13/B*35** | **DRB1*04:07:01/DRB1*07:01:01** | **DQB1*02:02:01/DQB1*03:01:01** | **DQA1*02:01/DQA1*03:03:01** |
| 7 | A*01/A*02 | B*08/B*08 | DRB1*11:01:01/DRB1*15:01:01 | DQB1*06:02:01/DQB1*03:01:01 | DQA1*01:02:01/DQA1*05:05:01 |
| 8 | A*02/A*33 | B*14/B*35 | DRB1*01:02:01/DRB1*12:01:01 | DQB1*03:01:01/DQB1*05:01:01 | DQA1*01:01:02/DQA1*05:05:01 |
| **9** | **A*02/A*11** | **B*35/B*44** | **DRB1*04:01:01/DRB1*08:01:02** | **DQB1*04:02:01/DQB1*03:01:01** | **DQA1*03:01:01/DQA1*04:01:01** |
| **10** | **A*02/A*25** | **B*44/B*18** | **DRB1*04:01:01/DRB1*14** | **DQB1*03:01:01/DQB1*05** | **DQA1*01:04:01/DQA1*03:03:01** |
| **11** | **A*02/A*03** | **B*07/B*50** | **DRB1*07/DRB1*07** | **DQB1*02/DQB1*02** | **–** |
| 12 | A*02/A*03 | B*35/B*58 | DRB1*01:01:01/DRB1*04:01:01 | DQB1*03:02:01/DQB1*05:01:01 | DQA1*01:05:01/DQA1*03:01:01 |
| **13** | **A*01/A*02** | **B*08/B*44** | **DRB1*03:01:01/DRB1*07:01:01** | **DQB1*02:01:01/DQB1*05:02:01** | **DQA1*02:01:01/DQA1*05:01:01** |
| **14** | **A*02/A*11** | **B*07/B*35** | **DRB1*07:01:01/DRB1*13:02:01** | **DQB1*03:03:02/DQB1*06:04:01** | **DQA1*01:02:01/DQA1*02:01:01** |
| **15** | **A*02/A*31** | **B*40/B*57** | **DRB1*04:01:01/DRB1*07:01:01** | **DQB1*03:01:01/DQB1*03:03:02** | **DQA1*02:01:01/DQA1*03:01:01** |

Bold: IgD⁺ NLPHL.

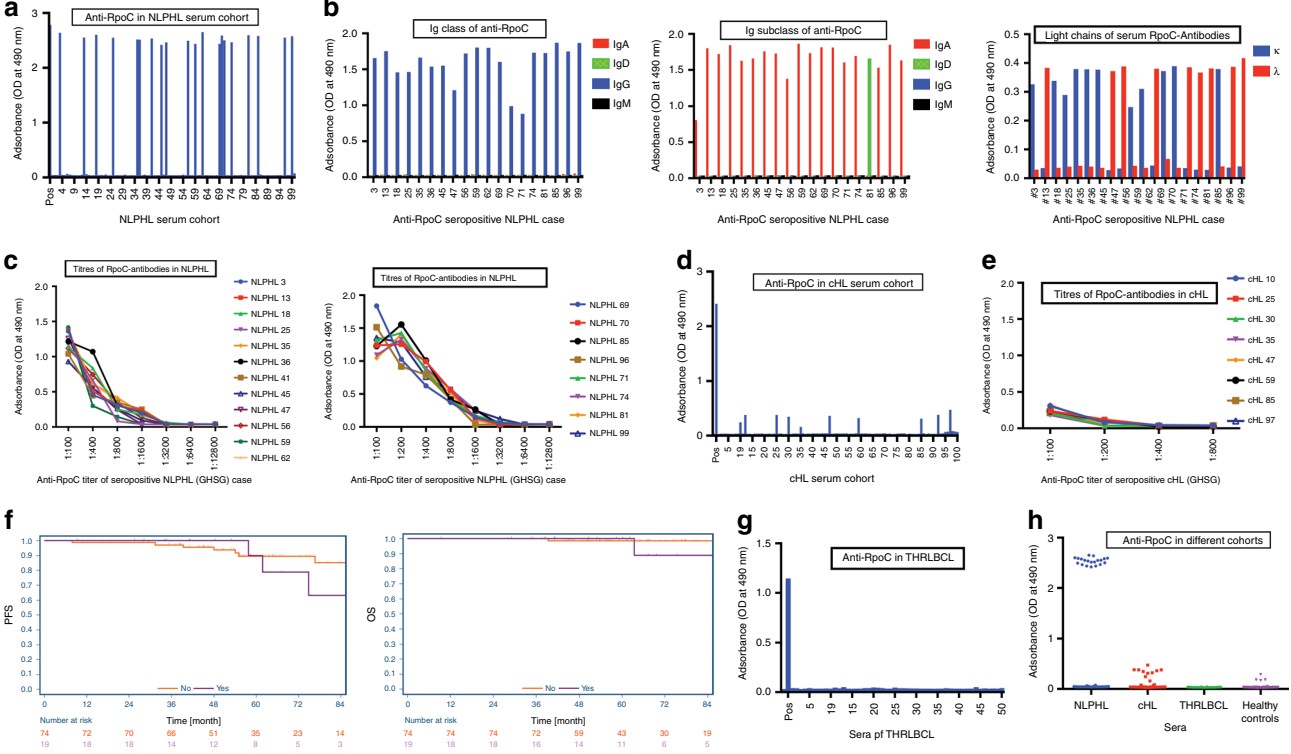

**Fig. 2 Serological responses against RpoC of *M. catarrhalis* in NLPHL, cHL, and THRBCL. a** Sera of patients with NLPHL (diluted 1:100) were tested for antibodies against RpoC of *M. catarrhalis*. **b** Analysis of Ig classes (left), IgG subclasses (middle), and light chains (right) of serum-antibodies against RpoC of *M. catarrhalis* in patients with NLPHL. The anti-RpoC antibodies were exclusively of the IgG class, mostly IgG1, and showed light chain restriction. **c** Titers of anti-RpoC antibodies in seropositive patients with NLPHL. The curves represent OD at 490 nm of different serum dilutions. Patients with NLPHL had titers between 1:800 and 1:3200. **d** Sera of patients with cHL (diluted 1:100) were tested for antibodies against RpoC of *M. catarrhalis*. **e** Seropositive patients with cHL had RpoC-antibody titers of up to 1:200. **f** PFS and OS of patients with NLPHL with and without RpoC-antibodies. No significant difference was observed for PFS ($p = 0.13$, Log-Rank-test) and OS ($p = 0.26$, Log-Rank-test). **g** Sera of patients with THRBCL (diluted 1:100) were tested for antibodies against RpoC, but all patients with THRBCL were seronegative for RpoC-antibodies. Serum of a seropositive NLPHL patient served as positive control. **h** Scatter blot for comparison of RpoC-antibodies in sera of patients with NLPHL, cHL, or THRBCL, and healthy controls. Data in **a** and **d** are representative of three independent experiments, data in **b**, **c**, **e**, **g**, and **h** are representative of two independent experiments.

RpoC-antibody-seropositive NLPHL patients ($n = 20$) did not differ in their clinical characteristics (age, gender, stage, site of manifestation, progression-free, and overall survival) from RpoC-antibody-seronegative NLPHL patients (Fig. 2f and data not shown). Unfortunately, the IgD-status was not available for this cohort. In the sera from 50 patients with THRLBCL, which represents a closely related entity to NLPHL, no RpoC-antibodies were found (Fig. 2g). Summarizing, anti-RpoC-antibodies were

frequently found in NLPHL patients, but not in patients with related lymphomas or healthy controls (Fig. 2h).

**RpoC and MID/hag induce additive stimulation**. Since we hypothesized that specific stimulation of the LP cell BCR by RpoC contributes to the lymphomagenesis of IgD+ NLPHL, patient-derived RpoC-specific BCRs were functionally investigated. At baseline, neither recombinant *M. catarrhalis* RpoC nor MID/hag resulted in increased proliferation of the IgD− DEV cell line, which is the only available NLPHL cell line (not shown). Similar results were observed for DEV cells stably expressing transfected BCRs without reactivity against RpoC. In contrast, RpoC stimulation resulted in the significantly increased proliferation of DEV cells that stably expressed RpoC-reactive IgD+-BCRs (Fig. 3a, b). Likewise, when MID/hag fragments 2/3 or 3 were applied by itself, already a significantly increased proliferation rate was observed (Supplementary Fig. 11). However, proliferation was further increased by the costimulation with RpoC and the MID/hag fragments 2 and 3, which contain the IgD-binding region (amino acids 920–1200) (Fig. 3a–c and Supplementary Fig. 11). This additive effect of MID/hag stimulation was not observed when DEV cells were transfected with an RpoC-reactive BCR of the IgG or IgM subtype (Fig. 3a, b). Accordingly, RpoC and MID/hag stimulation resulted in the strong activation of the BCR-signaling pathway, as determined by phosphorylation of key BCR-signaling factors (pTyr525/526 spleen tyrosine kinase [SYK], pTyr96 B-cell linker [BLNK], pTyr759 phosphoinositide-specific phospholipase C [PLC]γ2, and pTyr223 Bruton's tyrosine kinase [BTK]) in DEV cells that stably expressed RpoC-reactive IgD+ BCRs (Fig. 3c). The activation of the BCR pathway was associated with a significant increase in MYC expression (Fig. 3c). Additionally, the NF-κB pathway was activated after stimulation with RpoC and MID/hag (Supplementary Fig. 12). Activation was also shown by flow cytometric analysis of cytoplasmic calcium levels in DEV cells that stably expressed RpoC-reactive IgD+ BCRs by an increase in the intracellular calcium levels after incubation with *M. catarrhalis* RpoC, which could be further increased by coincubation with RpoC and MID/hag, but was not induced by a control antigen (MAZ, Supplementary Fig. 13).

**Antigen/drug conjugates target RpoC-reactive LP cells**. By using a synthetic FITC-conjugated RpoC peptide, which contained the binding epitope, the binding and internalization of RpoC into DEV cells that stably expressed RpoC-reactive BCRs could be measured using flow cytometry (Fig. 3d). In a LDH release assay, RpoC conjugated to a truncated form of *Pseudomonas aeruginosa* exotoxin A (RpoC/ETA'), was found to be cytotoxic against the DEV cell line that stably expressed RpoC-specific BCRs but had no effects on non-transfected DEV cells or on DEV cells that expressed BCRs specific for antigens other than RpoC (Fig. 3e). As shown by a trypan blue exclusion assay, 50% of the NLPHL cells expressing an RpoC-specific BCR died within 48 h after incubation with RpoC/ETA' (Supplementary Fig. 14). In support of these results, an increase in the number of apoptotic cells was detected by an AnnexinV/propidium iodide assay (Fig. 3f).

**Autoantigenic targets of BCRs derived from IgD− LP cells**. Two autoantigens, human ribosomal protein S27a (RPS27a, for NLPHL #1, AA1-AA95) and human pyruvate carboxylase (for NLPHL #5, AA1030-AA1178), were identified as antigenic targets of the BCRs of two individual IgD− NLPHL cases. Reactivity of the respective Fabs was confirmed by ELISA using recombinantly expressed RPS27 and pyruvate carboxylase with C-terminal FLAG-tag expressed in HEK293 cells (Supplementary Fig. 7A,

B). ELISAs with different fragments of pyruvate carboxylase revealed AA1111–AA1177 as the binding region of the recombinant Fab of patient #5 (Supplementary Fig. 7C). When comparing the identified antigens from the patients with autoreactive NLPHL-BCRs with those of healthy controls by Western-blot no obvious difference was detectable (Supplementary Fig. 7D). However, when comparing them by isoelectric focusing (IEF) a different electric charge of pyruvate carboxylase exclusively in patient #5 was observed (Supplementary Fig. 7E). This different electric charge was not altered after treatment with alkaline phosphatase (Supplementary Fig. 7F), but recombinant biotinidase pretreatment ahead of IEF resulted in the disappearance of the different electric charge, confirming a differential biotinylation of pyruvate carboxylase of patient #5 (Supplementary Fig. 7G). The hypobiotinylated pyruvate carboxylase might contribute to its immunogenicity in patient #5. Autoreactive BCRs contributing to lymphomagenesis by chronic stimulation due to alternative secondary modifications have previously been described in MGUS, Waldenström macroglobulinemia, multiple myeloma, and primary central nervous system lymphoma[19–22]. Pyruvate carboxylase is a highly conserved gene, and has been identified as an antigen recognized by BCRs of CLL[23].

## Discussion

Here, we describe a distinct subtype of NLPHL, with IgD+ LP cells, characteristically long IgVH CDR3s (median 30 amino acids), a high frequency of IgV gene mutations, and characteristic VDJ-recombination. IgD+ NLPHL previously defined a unique clinical NLPHL subset, which primarily affected the cervical lymph nodes of male adolescents[2,24]. Our results provide evidence that chronic antigenic simulation by a common bacterium contributes to lymphomagenesis in NLPHL[25]. Antigens produced by *M. catarrhalis* or *M. osloensis* were specifically recognized by recombinant BCRs derived from 7/8 NLPHL cases with IgD+ LP cells. Further analysis identified *M. catarrhalis* RpoC as the specific antigen for six *M. catarrhalis*-reactive BCRs and SUCLG1 as specific antigen for the single *M. osloensis*-reactive BCR. *M. catarrhalis*-derived RpoC bound specifically to the antigen-binding sites of IgD+ NLPHL-derived BCRs, which were specific for *Moraxella*, with no cross-reactivity against >200 pathogens, including various bacterial, fungal, parasitic, and viral pathogens. Likewise, BCRs from 45 B-NHLs, including DLBCL, primary central nervous system lymphoma, mantle cell lymphoma, and chronic lymphocytic leukemia, did neither bind to the *M. catarrhalis* lysate nor to RpoC. The presence of high-titer *M. catarrhalis*-RpoC antibodies in NLPHL serum samples suggests generally an adaptive humoral immune response against *M. catarrhalis*. The fact that these antibodies were predominantly of the IgG1 subclass and in addition light chain restricted, suggests a clonal B-cell expansion apart from the LP cell clone, that underwent class switch recombination and plasmocytic differentiation. These findings suggest that naïve IgD+ B cells become activated in an adaptive humoral immune response against *M. catarrhalis* and represent the precursors of LP cells.

*M. catarrhalis* is a widespread, Gram-negative bacterium that causes recurrent airway infections and otitis media[26]. *M. catarrhalis* additionally expresses the IgD-binding protein MID/hag, a bacterial autotransporter that binds specifically to the IgD Fc region and activates IgD+ B cells in a superantigenic manner[27–29]. Of interest, the outer membrane vesicles of *M. catarrhalis* contain both MID/hag and RpoC[17] and activate B cells[28], suggesting that *M. catarrhalis* can stimulate IgD+ LP cells in an additive manner, through RpoC binding to the antigen-binding region and through MID/hag binding to the Fc region of the BCR (Fig. 4). This result implies the combined antigenic and

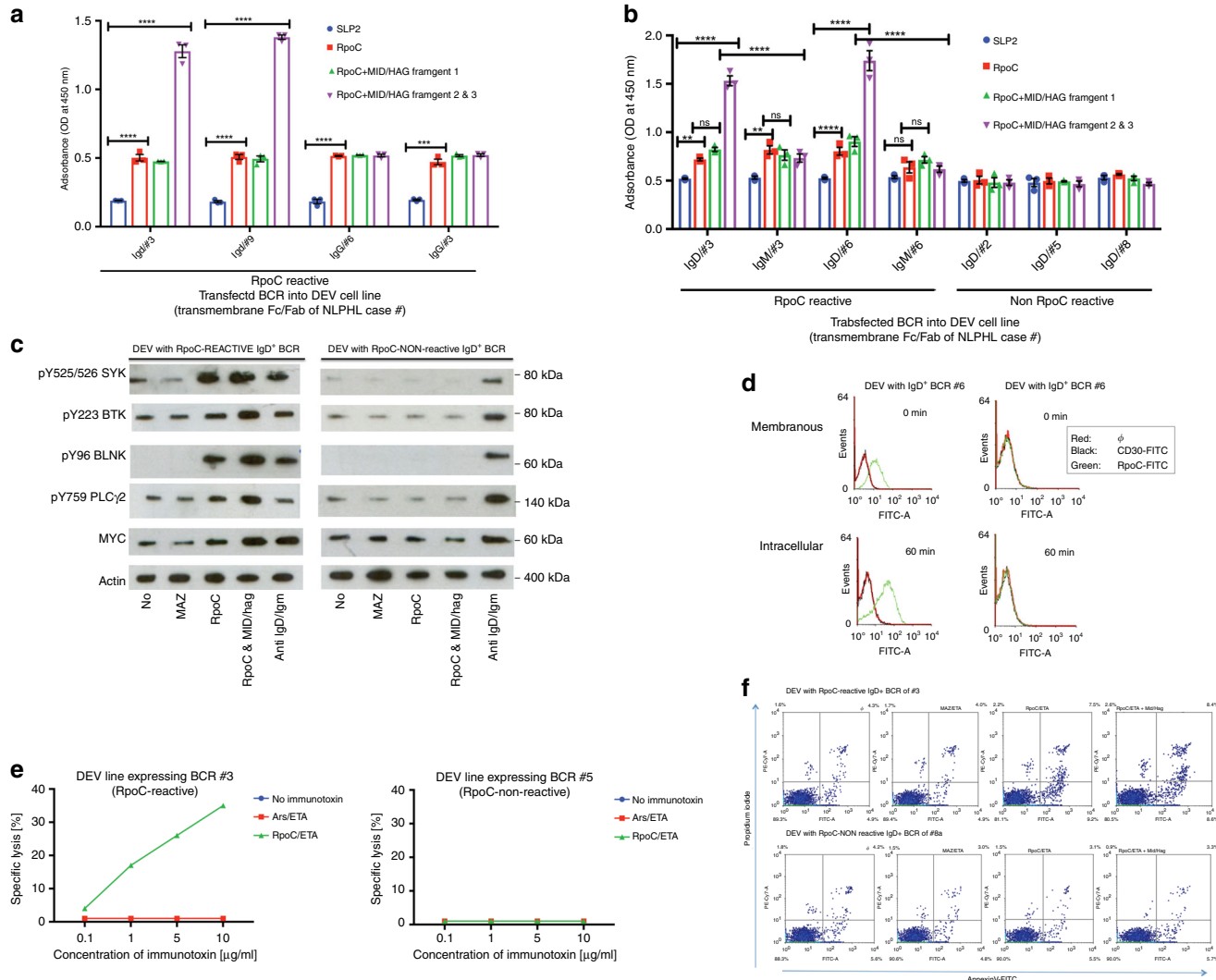

**Fig. 3 Costimulation of different types of LP cells by RpoC and MID/hag. a** Tetrazolium proliferation assay with the transfected DEV cells expressing recombinant BCRs with different RpoC-reactive Fab fragments and Fc fragments of either IgD or IgG class. Stimulation with RpoC resulted in significant proliferation, and costimulation with MID/hag fragments 2 and 3 resulted in strong further increase of proliferation in cell lines expressing BCR of IgD class, e.g. transfected DEV expressing IgD+ BCR #3 stimulated by rec. human SLP2 vs. rec. *M. catarrhalis* RpoC: $p < 0.00001$ (DF = 32, $q = 17.99$); DEV expressing IgD+ BCR #3 stimulated by rec. human SLP2 vs. rec. *M. catarrhalis* RpoC and MID/hag 2&3: $p < 0.00001$ (DF = 32, $q = 46.22$). **b** In a Tetrazolium assay stimulation with RpoC resulted in increase of proliferation in DEV cells expressing RpoC-reactive BCRs, but not in DEV cells expressing non-RpoC-reactive BCRs. In IgD+ DEV cells with RpoC-reactive BCR incubation with MID/hag 2 and 3 resulted in a significant increase of proliferation, whereas costimulation with MID/Hag fragments 2 and 3 and the control antigen SLP2 had no additive effect. DEV expressing IgD+ DEV cells with RpoC-reactive BCR incubation with MID/hag 2 and 3 resulted in a significant increase of proliferation, whereas costimulation with MID/Hag fragments 2 and 3 and the control antigen SLP2 had no additive effect. DEV expressing IgD+ BCR #6 stimulated by rec. human SLP2 vs. rec. *M. catarrhalis* RpoC: adjusted $p < 0.0001$ (DF = 56, $q = 7.321$) and DEV expressing IgD+ BCR #6 stimulated by rec. *M. catarrhalis* RpoC and MID/hag 2&3 vs. DEV expressing IgM+ BCR #6: $p < 0.00001$ (DF = 56, $q = 29.44$). Values in **a** and **b** are mean ± SEM. Two-way ANOVA with Tukey's multiple comparisons were performed, **$p \leq 0.01$, ***$p \leq 0.001$, ****$p \leq 0.0001$. Experiments were independently repeated three times. **c** Activation of the BCR-signaling pathway. Representative Western blot analysis of the BCR signaling pathway shows a strong activation by RpoC and even stronger by coincubation with RpoC and MID/hag in DEV cells transfected to express an RpoC-reactive IgD+ BCR. The incubation with RpoC, RpoC&Mid/Hag, or anti-IgM/IgD antibody results in the upregulation of the activated isoforms pTyr525/526 SYK, pTyr96 BLNK, pTyr759 PLCγ2, and pTyr223 BTK and induced overexpression of MYC. **d** Binding and internalization of RpoC/FITC into LP cells. RpoC/FITC bound to DEV cells transfected to express recombinant BCR of case #6 with RpoC-reactivity, but not to DEV cells transfected to express recombinant BCR of case #5. After 60 min internalization of RpoC/FITC into DEV cells with the RpoC-specific BCR #6 was observed. Control CD30/FITC neither bound to DEV cells nor was internalized. **e** Dose-dependant cytotoxic effects by ETA' toxin-conjugated RpoC dependent on expression of RpoC-reactive BCRs on LP cells. Specific cytotoxicity evoked by RpoC/ETA' was observed in the DEV cell line expressing the RpoC-reactive BCR of patient #3 (left), but not in the one expressing the autoreactive BCR of patient #5 (right). **f** Apoptosis induced by ETA' toxin-conjugated RpoC dependent on expression of RpoC-reactive BCRs on LP cells. Characterization of DEV cells stably transfected to express IgD+ recombinant BCRs with (above, from case #3) or without (below, from case #8a) reactivity to *M. catarrhalis* RpoC by Annexin-V/FITC and propidium iodide staining after 24 h cultivation in the presence of RpoC/ETA, MAZ/ETA, or RpoC/ETA together with MID/hag. RpoC/ETA induced specifically apoptosis in the DEV cell line with RpoC-reactive BCR. Data in **a**–**c** and **f** are representative of three independent experiments, data of **d** and **e** are representative of two independent experiments.

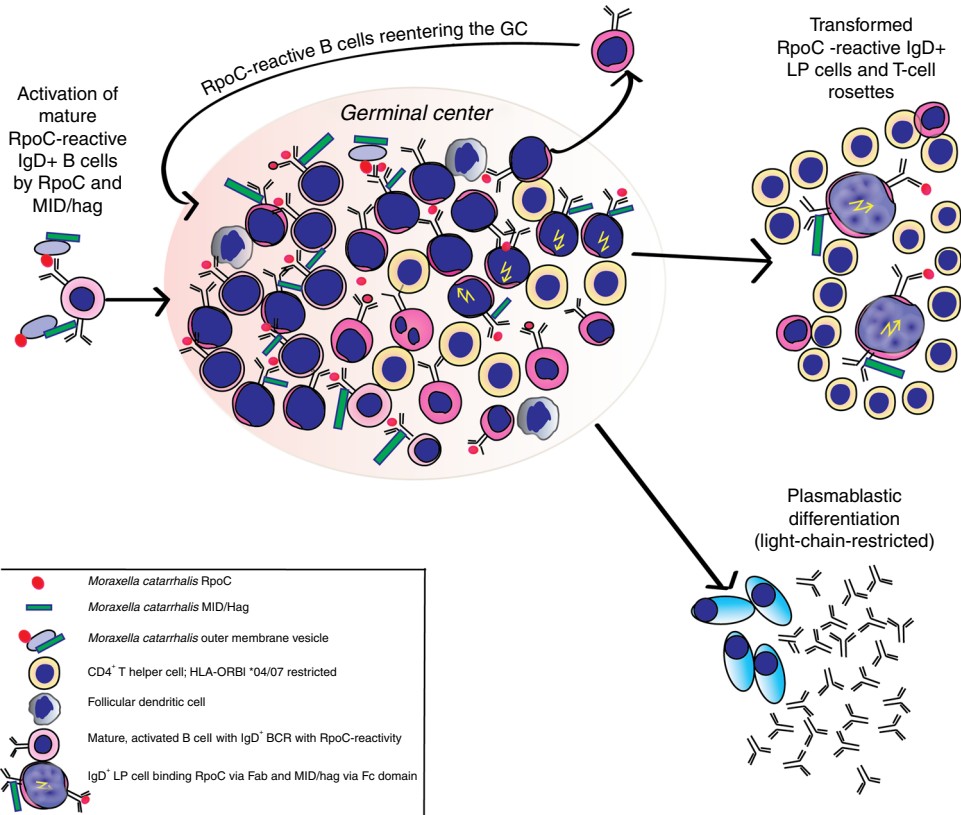

**Fig. 4 Presumable role of *M. catarrhalis* in the pathogenesis of IgD⁺ NLPHL.** Costimulation of B cells characterized by IgD⁺ BCR and extraordinary long CDR3s by *M. catarrhalis* RpoC via the Fab fragment and MID/hag via the Fc fragment of their BCR. Naïve, mature IgD⁺ B cells with a BCR specific for RpoC encounter *M. catarrhalis* outer membrane vesicles. Binding of RpoC to the Fab and of MID/hag to the Fc region of membranous IgD induces activation of RpoC-specific IgD⁺ B cells, which is supported by CD4⁺ T cells particularly in patients with an HLA-DRB1*04 or DRB1*07 haplotype. The persistent or recurrent presence of *M. catarrhalis* presumably induces a germinal center reaction resulting in differentiation to memory B cells and plasmablasts and production of class-switched anti-RpoC serum antibodies and apoptosis of some germinal center cells due to disadvantageous mutations. The co-stimulatory effect of MID/hag through binding to the Fc part of IgD selects for retention of IgD on a fraction of the stimulated germinal center B cells. Meanwhile a clone acquires mutations in proto-oncogenes, tumor suppressor genes as well as chromosomal translocations and may transform into LP cells. However, the transformed IgD⁺ LP cells still gain growth advantage by the interaction of their BCRs with *M. catarrhalis* RpoC and MID/hag.

superantigenic stimulation of RpoC-reactive BCRs, which is supported by our in vitro model using the DEV cell line to express IgD⁺ patient-derived RpoC-specific BCRs. The RpoC-induced increase in proliferation was only observed in DEV cells transfected with RpoC-reactive BCRs. The combination of RpoC and MID/hag fragments that contained the IgD-binding domain resulted in the strong and additive BCR pathway activation and proliferative effects in DEV cells that expressed RpoC-reactive IgD⁺ BCRs.

A potential role for superantigen-driven germinal center reactions in the generation of highly mutated IgV regions in so-called IgD-only B cells has previously been proposed[30]. The high mutation load in IgVH genes isolated from IgD⁺ NLPHL patients identifies a germinal center origin for the respective LP cells, which would require cognate T-cell help. This is supported by the presence of "permissive" HLA-DRB1*04/*07 haplotypes. Indeed, several high-affinity T-cell epitopes are predicted for these haplotypes, suggesting that these haplotypes can provide cognate follicular T-cell help for RpoC-specific B cells. Thus, in addition to the two *M. catarrhalis*-derived stimuli, a third stimulus may be provided in respective individuals by antigen-specific CD4⁺ T-helper cells, resulting in the extensive proliferation of B cells with a specificity for *M. catarrhalis* RpoC. During germinal center reactions, the chronically stimulated B cells likely acquire transforming events, resulting in the development of the malignant clone[4]. Because all RpoC-reactive BCRs derived from LP cells

were IgD⁺, the additional stimulation by the MID/hag protein is likely to be essential for the pathogenetic role played by *M. catarrhalis* in this scenario, as this finding indicates that the LP precursor cells are selected to retain IgD expression[31–33]. Contributions to lymphomagenesis from the chronic stimulation by T helper cells that have been activated by infectious pathogens or autoantigens have been demonstrated for some other lymphomas[32–34]. However, with the exception of hepatitis C virus, the target antigens of the BCRs from these lymphomas have either not been defined or have been shown to be autoantigens, and a combined antigen-specific and superantigenic stimulation, as has been identified here for NLPHL, has never been reported.

In contrast with NLPHL, no RpoC-antibodies were observed in the sera of a cohort of 50 THRLBCL patients. THRLBCL is an aggressive B-cell lymphoma that is closely related to NLPHL. Transformations of NLPHL into THRLBCL-like cases have been observed. However, the lack of RpoC-antibodies in THRLBCL patients suggests a different pathogenic mechanism than that observed for the *Moraxella*-induced NLPHL subtype. Furthermore, NLPHL cases that are *Moraxella* sp.-induced do not present with a THRLBCL-like morphology. Given recent reports from retrospective analyses regarding less intensive treatments and active surveillance strategies for NLPHL[35], the results of the present study might further raise discussions regarding how to treat this *Moraxella* sp.-induced NLPHL subgroup, in particular. These results raise the question of whether antibiotic treatment

and/or vaccination strategies could be used to prevent relapses of *Moraxella* sp.-induced NLPHL. Finally, the shared, specific BCR antigen found in IgD[+] LP cells could be therapeutically exploited as the targeting moiety for different therapeutic formats, such as antigen/toxin conjugates, bispecific CD3 or CD16 constructs or CAR-T cells, resulting in the selective delivery of a therapeutic payload to RpoC-reactive B cells and leaving other B cells unaffected[36].

## Methods

**Study samples.** The study was approved by the local ethics committees of the Universities in Frankfurt and Homburg. Frozen tissue sections of NLPHL specimens from the Dr. Senckenberg Institute of Pathology, Goethe University Hospital, Frankfurt/Main, Germany, from the Department of Pathology, Tampere University, Tampere, Finland, from the Institute of Pathology, CHUV Lausanne, Switzerland, and from the Department of Pathology, Uppsala University Hospital, Sweden, were analyzed. Informed consent of the patients was obtained in accordance with the Declaration of Helsinki. Sera of patients with NLPHL and sera of healthy controls were obtained at the medical school Homburg/Saar, Germany. Moreover, sera of patients with cHL or NLPHL were obtained from the HD13 (ISRCTN63474366), HD14 (ISRCTN04761296), HD15 (ISRCTN32443041), HD16 (NCT00736320), HD17 (NCT01356680) and HD18 (NCT00515554) trials of German Hodgkin Study Group (GHSG) as control and validation cohort[37–40]. Sera from patients with THRLBCL were obtained from the RICOVER (NCT00052936), Hi-CHOEP (NCT00129090), UNFOLDER (NCT00278408), FLYER (NCT00278421), CHOP-R-ESC (NCT00290667) trials of the German High-Grade Non-Hodgkin Lymphoma Study Group (DSHNHL)[41–44].

**Laser microdissection and IgV region gene PCR.** Single, clearly identifiable LP cells were microdissected with an ultraviolet laser (PALM microdissection system, Zeiss Axiovert 200M microscope), pooled in groups of 30 cells and resuspended in 18 µl 1x PCR buffer and digested with 2 µl of proteinase K (Roche, Grenzach, Germany) at 55 °C for 4 h, followed by enzyme inactivation at 95 °C for 10 min. The LP cell lysates were subjected to two rounds of $V_H$-, $V_\kappa$-, and $V_\lambda$-specific seminested PCRs (30 and 44 cycles in the first and second round of PCR, respectively) using IgV family-specific primers and J primer mixes, and Expand high fidelity PCR kit (Roche) as described by Küppers et al.[45].

**Expression of recombinant BCRs.** The amplified IgV region genes were sequenced and analyzed with IMGT-V-Quest for functionality, V, D, and J segment usage and indications for somatic mutations. If both a functional heavy and light chain variable region gene was amplified, the IgV region genes were cloned into TOPO Zero-Blunt vector (Invitrogen Life Technologies, Darmstadt, Germany)[46]. IgV gene fragments were re-extended at the 5′ and 3′ ends according to the proper immunoglobulin germline genes. Complete IgV genes were inserted via ApaLI and XhoI for $IgV_\kappa$ or $IgV_\lambda$ in front of a κ-constant or λ-constant region gene, respectively, and via NcoI and BstEII for $IgV_H$ in front of a γ-1 constant region gene into a modified pCES-1 vector for expression of the Fab fragments[47]. Fabs were expressed and purified[23].

**Antigen screening.** To screen for potential reactivity of Fabs against bacterial antigens, heat-inactivated lysates of 13 different bacterial strains or patient isolates including *M. catarrhalis*, *M. osloensis*, *M. nonliquefaciens*, *Streptococcus pneumoniae*, *Streptococcus pyogenes*, *Haemophilus influenzae*, *Staphylococcus aureus*, coagulase-negative staphylococci, *Klebsiella pneumoniae*, *Enterobacteriacae*, *Escherichia coli*, *Acinetobacter radioresistens*, *Neisseria* spp., as well as DNA from *M. catarrhalis* and *osloensis* isolated from a patient, were provided by the Institute of Medical Microbiology and Hygiene of the University of Saarland, Homburg/Saar, Germany. Inactivated lysates of *M. catarrhalis* strain ATCC 43617 RO 108 and lysates of *Bartonella henselae* strain Marseille (*Bartonella* adhesin A positive), were provided by the Institute of Medical Microbiology and Infection Control of the Hospital of the Goethe University Hospital, Frankfurt/Main (Supplementary Table 3), were spotted onto PVDF membranes each with a dose of 10 µg. The dot blots of bacterial lysates were blocked in 10% (w/v) non-fat dry milk powder in TBST [TBS, 0.1% (v/v) Tween 20] at 4 °C overnight, washed twice in TBST and incubated for 1 h with the individual Fabs, each at a concentration of 10 µg/ml. Following three 30-min TBST washes and subsequent incubation for 1 h at room temperature with biotinylated goat anti-human heavy and light chain Fab antibody (DIANOVA, 109-065-088) at a dilution of 1:5000 (v/v) the arrays and blots were incubated for 10 min at room temperature with Strep-POX (1:5000) in 2% (w/v) milk/TBST and binding was detected using the ECL system (Amersham Pharmacia, Freiburg, Germany). Unprocessed scans of the dot blots are provided at the end of the Supplementary information.

To extend the screening for infectious antigens, the recombinant NLPHL-Fabs were screened together with recombinant Fabs of different B-cell neoplasias at concentrations of 1, 10, and 20 µg/ml on an infectious disease epitope microarray (PEPperCHIP®/Heidelberg, Germany) consisting of 3760 database-derived B-cell

epitopes from 190 pathogens, including 113 viruses, 41 bacteria, 25 parasites, 10 worms, and 1 pathogenic fungus. Goat anti-human IgG (H + L) conjugated to DyLight680 (1:5000) was used as secondary antibody and incubated for 45 min at room temperature. As a scanner the LI-COR Odyssey Imaging System was used with a scanning offset of 0.65 mm, a resolution of 21 µm and scanning intensities of 7/7 (red = 700 nm/green = 800 nm). Data quantification was followed by removal of spots with a deviation of more than 40%. Screening, scanning, and data analysis was performed by PEPperPRINT GmbH in Heidelberg, Germany.

To screen Fabs against autoantigens, high-density cDNA library protein expression macroarrays (UniPex 1 & 2, Bioscience, Dublin, Ireland) consisting of 15,300 spotted UniPEx expression clones derived from human fetal brain, T cells, lung, and colon, which, after induction of expression, represent 7390 distinct recombinant human proteins were used[19]. Protein-macroarrays were blocked in 10% (w/v) non-fat dry milk powder in TBST at 4 °C overnight, washed twice in TBST and incubated for 1 h with pooled Fabs for the protein-arrays, each at a concentration of 10 µg/ml. Following three 30 min TBST washes and subsequent incubation for 1 h at room temperature with biotinylated goat anti-human Fab antibody (DIANOVA, 109-065-088) at a dilution of 1:5000 (v/v) the arrays and blots were incubated for 10 min at room temperature with Strep-POX (1:5000) in 2% (w/v) milk/TBST and binding was detected using the ECL system (Amersham Pharmacia).

**Antigen characterization.** To characterize antigens detected in the *M. catarrhalis* lysate, PAGE separations of *M. catarrhalis* lysates from a clinical isolate propagated from a patient with respiratory infection and a lysate of the strain ATCC 43617 RO 108 were performed and were Western blotted under reducing and non-reducing conditions with *M. catarrhalis*-reactive LP cell-derived recombinant Fabs and highly diluted (10−6) serum of a patient with IgD-secreting multiple myeloma. To identify the specific *M. catarrhalis* antigen of 150–160 kDa detected by Western blot out of more than 1800 known *M. catarrhalis* genes, an approach based on published literature and hypothetic assumptions was chosen to select possible candidate antigens. Literature search resulted firstly in the selection of MID/hag as possible candidate antigen, as it had previously been proposed as an antigen involved in the generation of IgD-only B cells, although the molecular weight is reported to be higher than that of the identified target antigen[30]. Moreover, only very few proteins of *M. catarrhalis* outer membrane vesicles have been reported to have as high molecular weights as the identified antigen of 150–160 kDa[16]. In a published list of the 50 most immunogenic B cell epitopes of *M. catarrhalis*, RpoC was the only protein with a molecular weight around 155 kDa[48]. By these means RpoC was selected as second hypothetic *M. catarrhalis* candidate antigen.

To characterize antigens detected in the *M. osloensis* lysate, the lysate was separated in two-dimensional gels, followed by silver staining and Western blot with *M. osloensis*-reactive LP cell-derived recombinant Fabs as primary antibody (patient #11) followed by mass spectroscopy (Proteome Factory AG, Berlin, Germany). In short, NCBIprot database version 20161128 was used (106762850 sequences, 39119668168 residues) with the following parameters: enzyme: trypsin; fixed modifications: carbamidomethyl (C); variable modifications: deamidated (NQ), oxidation (M); mass values: monoisotopic; protein mass: unrestricted; peptide mass tolerance: ±3 ppm; fragment mass tolerance: ±0.6 Da; max missed cleavages: 1; instrument type: ESI-Trap.

The MID/hag and RpoC gene from *M. catarrhalis* and SUCLG1 gene from *M. osloensis* were amplified from the respective bacterial DNA. The expression clones of human pyruvate carboxylase, RPS27a and SMCHD1 were obtained from the manufacturer (Bioscience, Dublin, Ireland), the latter as control antigen, as reported previously[23]. Genes of MID/hag and RpoC from *M. catarrhalis* and of SUCLG1 from *M. osloensis* were cloned (Supplementary Table 1) with a C-terminal FLAG-tag into pSFI vector and expressed in HEK293. Nunc maxisorb ELISA plates were coated overnight at 4 °C with murine anti-FLAG antibody (Sigma, F3165, Munich) 1:2500 v/v. After blocking with 1.5% (w/v) gelatin in TBS and washing steps with TBS with Triton-X, recombinant NLPHL-derived Fabs were added at 10 µg/ml or patient sera at 1:100 for 1 h at room temperature. After washing-step with TBS-Tx biotinylated goat anti-human IgG (heavy and light chain) (Dianova) 1:2500 v/v or sheep anti-human IgG1, IgG2, IgG3, and IgG4 (Binding Site, AU006, AU007, AU008, and AU009, Birmingham, UK) 1:5000 (v/v) or rabbit anti-human IgM (Dianova, 109-476-129) 1:2500 v/v were added for 1 h at room temperature followed by washing step with TBS. For the determination of IgG subclasses and IgM biotinylated correspondent secondary antibodies were applied. This was followed by peroxidase-labeled streptavidin (Roche) 1:50,000.

To verify specific reactivity of IgD[+], NLPHL derived Fabs against *M. catarrhalis* RpoC, recombinant Fabs derived from 11 cases of PCNSL, 12 cases of DLBCL, 10 cases of MCL, and 12 cases of CLL were screened against *M. catarrhalis* lysates on dot blots and against recombinant FLAG-tagged *M. catarrhalis* RpoC by ELISA[22,23,49].

**Generation of DEV cells expressing recombinant BCRs.** The only existing NLPHL cell line (DEV) was cultured at 37 °C in an atmosphere containing 5% $CO_2$ in RPMI-1640 medium supplemented with penicillin (100 U/ml), streptomycin (0.1 mg/ml), ultraglutamine (2 mM), and 20% fetal calf serum, and transfected with a pRTS expression vector with an IgV region heavy chain and constant regions Cγ1–Cγ4, Cµ1–Cµ4, or Cδ1–Cδ4 with membrane coding exons TM1 and TM2 for

the transmembrane region and the cytoplasmic tail followed by a 2A sequence and the light chain variable region and light chain constant region gene. Transfection of DEV cells was performed after three washing steps with RPMI-1640 at a cell density of $2 \times 10^7$/ml in RPMI-1640 without FCS on ice. Two times $10^6$ cells equalizing a volume of 100 µl were transfected with 5 µg plasmid DNA by electroporation using Gene Pulser (Biorad) with a 0.2 cm cuvette, a voltage of 140 V and 30 ms pulses. Subsequently cells were immediately put again on ice and cultured in RPMI-1640 medium supplemented with 20% FCS (Sigma, F2442). Cell lines stably expressing recombinant membranous BCR were selected with hygromycin at 250 µg/ml. Expression of recombinant BCR was induced by addition of doxycycline[50]. Successful transfection was verified by IgV region gene PCRs of transfected cell lines, by Western blot of the FLAG-tagged recombinant BCRs and surface expression of the transfected His-tagged BCR was determined by flow cytometry. In detail, surface expression of recombinant patient-derived BCRs on the membrane of DEV cell line was verified by flow cytometry using the BD FACS Canto and either the original DEV cell line or DEV cell lines transfected to express LP cell-derived BCR, which contain His6 tags. Cells were incubated with murine Anti-His antibody (Qiagen, 34660, Hilden, Germany) (1:500) followed by biotinylated anti-murine antibody (1:200) and PE-labeled streptavidin (Qiagen, 016-110-084) (1:500) each for 20 min at 4 °C with washing steps in between (Supplementary Fig. 15).

**BCR and NF-κB pathway activation and proliferation assay.** For Western blot analysis of the BCR pathway activation of transfected DEV cells either expressing a BCR with reactivity against *M. catarrhalis* RpoC or a different antigen, respectively, and of IgD, IgG, or IgM class, $1 \times 10^6$ cells were incubated with no antigen, recombinant RpoC at 5 µg/ml, RpoC and MID/hag fragment 2&3, MAZ at 5 µg/ml or anti-IgM/IgD at 1 µg/ml. As primary antibodies rabbit antibodies against pTyr525/526 SYK diluted 1:2000, pTyr759 PLCγ2 diluted 1:1000, pTyr223 BTK diluted 1:1000, and pTyr96 BLNK diluted 1:1000 (B cell signaling sampler kit, 9768, CST, Massachusetts, USA), rabbit antibody against actin diluted 1:2000 (Sigma, A5060), and murine antibody against MYC at a concentration of 1 µg/ml (Santa Cruz) were utilized, followed by washing steps and incubation with POX-conjugated anti-rabbit or anti-mouse antibodies diluted at 1:3000. For analysis of the NF-κB pathway mouse antibody against IKKα and IκBα, rabbit antibodies against IKK, pSer176/180 IKKα/β, pSer536 NF-κB p65, NF-κB p65, and against pSer32 IκBα were used (NF-κB pathway sampler kit, 9936, CST, Massachusetts, USA).

For the analysis of cytoplasmatic calcium changes by flow cytometry a FACS Canto analyzer was used and Fluo-4/AM dye (Molecular probes, Invitrogen, F14201). Transfected DEV cells either expressing IgD$^+$ BCRs with reactivity against RpoC or against a different antigen were resuspended in calcium-free and magnesium-free phosphate-buffered saline, and loaded with Fluo-4/AM dye (final concentration 2 µM, Invitrogen, Karlsruhe, Germany) for 30 min at room temperature. Antigen was added followed by flow cytometry of the cells. Ionomycin (10 ng/µl, Sigma-Aldrich, 407952) was used as a positive control for the release of calcium from internal stores. Intracellular calcium levels were repeatedly analyzed immediately after adding the antigen to the dye-loaded cells and mixing. To exclude cell debris DEV cell line was gated for relatively high FSC and low SSC.

EZ4U, a non-radioactive proliferation assay (Biomedica, BI-5000), was performed according to the manufacturer's instructions with DEV cells (IgD$^-$), and the DEV cell line transfected to express RpoC-reactive surface BCR of the IgD class, and as further control a DEV cell line expressing RpoC-reactive membranous IgG BCR. The IgV genes were derived from case 3, and the IgD constant region gene from PBMCs of a healthy blood donor[51,52]. *M. catarrhalis* RpoC and as a control recombinant human SLP2, which is a frequent antigenic target of paraproteins from patients with multiple myeloma, were added at 10 µg/ml. Recombinant fragments of *M. catarrhalis* MID/hag spanning amino acids 1–761 (fragment 1), amino acids 920–1368 (fragment 2), and amino acids 920–2090 (fragment 3) were added to DEV cell cultures at 5 µg/ml. Absorbance of Formazan at 450 nm was determined after 3 days at 37 °C. Statistical significance was calculated by multiple *t*-tests (Prism7, graphpad).

**Cytotoxicity and apoptosis assays.** For the analysis of the direct cytotoxic effects of immunotoxins a lactate dehydrogenase (LDH) release assay was used (Roche, 04744926001). $5 \times 10^3$ DEV cells per well, stably transfected to express the BCR of case #3 with reactivity against *M. catarrhalis* RpoC or of case #5 without reactivity against *M. catarrhalis* RpoC were incubated with RpoC/ETA or Ars2/ETA at concentrations from 0.1 to 10 µg/ml, or without an immunotoxin. Percent-specific lysis was determined after 24 h at 37 °C as follows: (experimental lysis−spontaneous lysis)/(maximum lysis–spontaneous lysis) × 100. Maximum lysis was determined by adding 10% Triton X-100. LDH was measured according to the protocol of the LDH assay kit (Roche, Mannheim, Germany). ELISA read out was done using a Victor II apparatus (PerkinElmer, Rodgau, Germany). For the analysis of apoptosis, $5 \times 10^5$ cells/ml suspension of DEV cells stably transfected to express either a BCR with reactivity against RpoC or a different antigen were treated by adding RpoC/ETA, MAZ/ETA (both at 0.5 µg/ml) or RpoC/ETA and MID/hag for 24 h at 37 °C, 5% $CO_2$. Following the incubation, cells were washed twice with PBS and resuspended in 500 µl binding buffer. 5 µl of AnnexinV-FITC (SIGMA, APOAF) and 10 µl of propidium iodide (SIGMA) were added to each cell

suspension and incubated for 10 min at room temperature, followed by analysis by FACS Canto. In addition, the effects of the immunotoxins were measured by trypan blue assays at 0, 24, and 48 h.

**HLA typing.** HLA-typing of class I and II human leukocyte antigens (HLA) was performed for all patients with IgD$^+$ LP cells by sequence-based typing (Labor Thiele, Kaiserslautern, Germany).

**Analysis of binding and internalization.** To analyze binding to and internalization of RpoC into DEV cells transfected to express RpoC-reactive BCRs, a synthetic RpoC peptide consisting of the amino acid sequence VAAKDVVNADGDVV VPSGALIDERL with a C-terminal FITC (purity > 95%, Genecust, Luxembourg) was added at a concentration of 10 µg/ml or as a control CD30/FITC to DEV lines transfected either with RpoC-reactive BCR (derived from case #6) or without RpoC-reactive BCRs (derived from case #5) for 30 min at 4 °C. Cells were then immediately, or after 1 h incubation at 37 °C analyzed. To determine surface binding or internalization of RpoC/FITC half of the cells were washed and analyzed directly by flow, and the other half of cells was washed, treated by azide 2% for 5 min to solve the surface-bound antigen, washed again followed by flow cytometry.

## Data availability

All relevant data are available in the Article, Supplementary Information or from the corresponding author upon reasonable request.

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

## Acknowledgements

This article is dedicated to the memory of M. Pfreundschuh, who died during its preparation. We are grateful to Patricia Flüchter, Sabine Albrecht, Ralf Lieberz, and Wibke Ballhorn for technical assistance. We thank Gabi Carbon, Claudia Schormann, Moritz Bewarder, and the DSHNHL team, Karola Lehmann, Andreas Meinke, Gerhard Held, Norbert Graf, and Bernhard Thurner for constant support, advice, and critical reading of the manuscript. Sources of Funding: M.P., L.T. K.-D.P., and F.N. have been supported by the Wilhelm Sander Stiftung. L.T. was supported by a University of Saarland fellowship. S.H. is supported by the Deutsche Forschungsgemeinschaft (grant HA 6145/3-1). V.A.J.K was supported by the Viral and Bacterial Adhesin Network Training (ViBrANT) Program funded by the European Union's HORIZON 2020 Research and Innovation Program under the Marie Sklodowska-Curie Grant Agreement No. 765042, by the Deutsche Forschungsgemeinschaft [DFG FOR 2251] and by the Robert Koch-Institute, Berlin, Germany (Bartonella consiliary laboratory, 1369-354).

## Author contributions

L.T., S.H.: design of study, performing experiments, data analysis, and writing manuscript; N.F., E.G., M.K., M.A.W., F.N., N.S.: performing experiments and data analysis; Y.-J.K., R.M.B., A.N., L.v.M., V.A.J.K., M.V., C.S., L.d.L., A.E., D.A.E: supplied essential material, interpretation of data, R.K., K.-D.P., M.-L.H., M.P.: design of study, interpretation of data, and revision of manuscript.

## Competing interests

The authors declare no competing interests.
