## [Peer Review File · Nature Communications]

This manuscript has been previously reviewed at another journal that is not operating a transparent peer review scheme. This document only contains reviewer comments and rebuttal letters for versions considered at Nature Communications.

Reviewers' comments:

Reviewer #1 (Remarks to the Author): expertise in B cells, antibodies, germinal centre response

The authors have improved the layout of their manuscript and addressed most of my primary points. A few smaller points remain that can be addressed editorially:

1. The figures are still hard to read with extremely small fonts and lacking important information in certain panels. For example, one shouldn't have to read the legend to find out what is depicted on a graph axis.

2. Pg 6: What exactly do the authors mean when they state that the serum responses are "light-chain restricted"? Fig 3b. shows both kappa and lambda light chains react with RpoC in serum in different patients. The either/or nature of these reactivities probably suggests these are monoclonal reactivities derived from the lymphoma via class-switching, and would of course express only one light chain. Unless I'm misunderstanding what the authors mean by "light-chain restricted" I wouldn't say this feature is worth highlighting, except maybe as evidence that the serum antibody must be derived from the malignant cells.

3. Figure 3d: "After 60 minutes internalization of 21 RpoC/FITC into DEV cells with the RpoC-specific BCR #6 was observed." How do the authors distinguish between intracellular and extracellular RpoC in the bottom panel?

4. Typo: Fig S11 should read "Moraxella" not "Moracella";

5. Supplementary figs. 13 and 14 are not mentioned in the text.

6. "The identification of light-chain-restricted, high-titer, *M. catarrhalis*-RpoC antibodies of the IgG1 subclass, with a specificity for the same RpoC epitope (Figure 1), in serum samples from two patients with RpoC-specific IgD + NLPHL and 20% of all GHSG NLPHL patients analyzed suggests an adaptive humoral immune response against *M. catarrhalis* RpoC, that includes class switching and plasmablastic differentiation. These findings suggest that activated IgD + B cells are involved in an adaptive humoral immune response against *M. catarrhalis* and represent the precursors of LP cells." This section is a bit confusing, see point #2 above.

Reviewer #2 (Remarks to the Author): expertise in B cells, Ab repertoire, germinal centre

The authors conclude that RpoC and MID/Hag had synergistic effects on DEV cells expressing RpoC-reactive BCRs. The previous reviewer comments raised the concern that this conclusion cannot be made without evaluating the effect of RpoC and MID/Hag alone--independent of each other in addition to together. The original manuscript had RpoC alone compared to RpoC+MID/Hag together. Despite the reviewer request, the addition of an MID/Hag alone control was not added to the revised manuscript. Wouldn't addition of a MID/Hag alone control be important as MID/Hag (fragments 2 and 3) by itself could be responsible for the increased activation seen?

Reviewer #4 (Remarks to the Author): expertise in lymphoma biology and treatment

Thurner et al. present an interesting study of IgD+ NLPHL, demonstrating specificity of BCR's in those cases to RpoC antigen from *M. catarrhalis*, further enhanced reactivity to the MID/hag superantigen. The findings were first preliminarily reported in 2015, appear to be specific to NLPHL, and the study is further strengthened by the RpoC-specific BCR being potentially targetable with a ligand-toxin conjugate.

This study is very novel, as it describes a pathophysiologic connection between a bacterial

infection and NLPHL—a rare and peculiar lymphoma characterized by extensive non-malignant lymphocytic infiltrate. Prior studies have shown that IgD+ NLPHL occurs predominantly in men and in cervical lymph nodes, which was unexplained. The current study shows that the connection with *Moraxella* is disease-specific, as IgD-negative NLPHL and various other lymphomas did not have reactivity with *Moraxella*. This is very exciting, as the identification of infectious agents as targets of lymphoma-derived BCR's/IgG has been elusive.

The authors have responded adequately to prior review, restructuring the paper and providing additional data. The residual weakness of lack of functional assessment in in vivo models remains (only a single existing NLPHL cell line was investigated), but can be resolved with future studies if samples of the rare IgD+ NLPHL are obtained prospectively. The specific requests for error bars in graphs and a Western blot for NFkB were met. It remains unclear whether patients with RpoC-reactive NLPHL had persistent infection with *M. catarrhalis* (and thus persistent source of the MID/hag superantigen) or not. Minor comments:

1. Text should refer to specific panels on figures rather than to the entire figure repeatedly.
2. It appears that the IgD+NLPHL which is associated with *M. catarrhalis* has specific clinicopathologic findings. However, in the second GHSG cohort, the authors find no association between high-titer RpoC reactivity and these findings. Is there an association between seropositivity and IgD+ status in that cohort? If these data are not available, this is a limitation of this study.
3. Fig. 3: addition of a control stimulated by MID/hag alone would be persuasive, as RpoC and MID/hag are always present together in vivo, so the principal effect on the IgD+ NLPHL cells could be through MID/hag.
4. P9: I would not describe IgD+NLPHL as having “a high mutational burden” which usually indicates a genomic mutation rate; the authors presumably refer to the rate of somatic hypermutation.

Point by point response to the issues raised by the referees

We thank the editor and the reviewers for very helpful comments and suggestions. We have revised our manuscript accordingly. Changes in the manuscript are highlighted in yellow.

Outline of specific responses to comments (comments black, > responses red):

Reviewer #1 (Remarks to the Author): expertise in B cells, antibodies, germinal centre response

The authors have improved the layout of their manuscript and addressed most of my primary points. A few smaller points remain that can be addressed editorially:

1. The figures are still hard to read with extremely small fonts and lacking important information in certain panels. For example, one shouldn't have to read the legend to find out what is depicted on a graph axis.

-> We agree with the reviewer and thank the reviewer for his/her comments. We have reorganized the figures, and parts of the figures and fonts were increased in size.

2. Pg 6: What exactly do the authors mean when they state that the serum responses are "light-chain restricted"? Fig 3b. shows both kappa and lambda light chains react with RpoC in serum in different patients. The either/or nature of these reactivities probably suggests these are monoclonal reactivities derived from the lymphoma via class-switching, and would of course express only one light chain. Unless I'm misunderstanding what the authors mean by "light-chain restricted" I wouldn't say this feature is worth highlighting, except maybe as evidence that the serum antibody must be derived from the malignant cells.

-> We apologize that it was not clearly explained in the manuscript that RpoC serum antibodies were of IgG class and thus probably not directly related to the LP cell clone. However, in all patients they presented light chain restriction to either kappa or lambda, suggesting that they resulted from a strong clonal stimulation of a respective RpoC-reactive clone. We have now added a better explanation to the respective paragraph on page 6.

3. Figure 3d: "After 60 minutes internalization of 21 RpoC/FITC into DEV cells with the

RpoC-specific BCR #6 was observed.” How do the authors distinguish between intracellular and extracellular RpoC in the bottom panel?

-> To determine surface binding or internalization of RpoC/FITC half of the cells were washed and analyzed directly by flow, and the other half of cells was washed, treated by 2% azide for 5 min to remove the surface bound antigen, washed again followed by flow cytometry. This information is now included in the supplementary information.

4. Typo: Fig S11 should read “Moraxella” not “Moracella”;

-> We thank the reviewer for this hint. The typo was corrected.

5. Supplementary figs. 13 and 14 are not mentioned in the text.

-> Supplementary figures 13 and 14 are now mentioned in the text. However, the order of the supplementary figures changed and one additional supplementary Figure was added (supplementary Figure 11 showing the stimulation of DEV cells with MID/hag fragments with or without RpoC). So, the former supplementary Figure 13 was changed to supplementary figure 15, which is mentioned in the Methods section and former supplementary figure 14 was changed to supplementary figure 7, which is mentioned in the Results section.

6. “The identification of light-chain-restricted, high-titer, M. catarrhalis-RpoC antibodies of the IgG1 subclass, with a specificity for the same RpoC epitope (Figure 1), in serum samples from two patients with RpoC-specific IgD+ NLPHL and 20% of all GHSG NLPHL patients analyzed suggests an adaptive humoral immune response against M. catarrhalis RpoC, that includes class switching and plasmablastic differentiation. These findings suggest that activated IgD+ B cells are involved in an adaptive humoral immune response against M. catarrhalis and represent the precursors of LP cells.” This section is a bit confusing, see point #2 above.

-> We agree with the reviewer that the wording of this part of the Discussion section is suboptimal. We have rephrased this part of the Discussion on pages 9 and 10.

Reviewer #2 (Remarks to the Author): expertise in B cells, Ab repertoire, germinal centre

The authors conclude that RpoC and MID/Hag had synergistic effects on DEV cells expressing RpoC-reactive BCRs. The previous reviewer comments raised the concern that

this conclusion cannot be made without evaluating the effect of RpoC and MID/Hag alone-- independent of each other in addition to together. The original manuscript had RpoC alone compared to RpoC+MID/Hag together. Despite the reviewer request, the addition of an MID/Hag alone control was not added to the revised manuscript. Wouldn't addition of a MID/Hag alone control be important as MID/Hag (fragments 2 and 3) by itself could be responsible for the increased activation seen?

-> We thank the reviewer for this important comment. We have added the results of the stimulation by different fragments of MID/Hag itself with and without RpoC as supplementary Figure 11. As the reviewer assumed, fragments 2/3 and fragment 3 of MID/Hag alone were sufficient to induce a significant proliferation in DEV cells transfected with an IgD-positive BCR. However, addition of RpoC together with fragments 2/3 or 3 of MID/Hag could still significantly increase the proliferation rate when compared with MID/Hag alone. This information has also been added to the Results section on page 8.

Reviewer #4 (Remarks to the Author): expertise in lymphoma biology and treatment

Thurner et al. present an interesting study of IgD+ NLPHL, demonstrating specificity of BCR's in those cases to RpoC antigen from *M. catarrhalis*, further enhanced reactivity to the MID/hag superantigen. The findings were first preliminarily reported in 2015, appear to be specific to NLPHL, and the study is further strengthened by the RpoC-specific BCR being potentially targetable with a ligand-toxin conjugate. This study is very novel, as it describes a pathophysiologic connection between a bacterial infection and NLPHL—a rare and peculiar lymphoma characterized by extensive non-malignant lymphocytic infiltrate. Prior studies have shown that IgD+ NLPHL occurs predominantly in men and in cervical lymph nodes, which was unexplained. The current study shows that the connection with *Moraxella* is disease-specific, as IgD-negative NLPHL and various other lymphomas did not have reactivity with *Moraxella*. This is very exciting, as the identification of infectious agents as targets of lymphoma-derived BCR's/IgG has been elusive. The authors have responded adequately to prior review, restructuring the paper and providing additional data. The residual weakness of lack of functional assessment in in vivo models remains (only a single existing NLPHL cell line was investigated), but can be resolved with future studies if samples of the rare IgD+ NLPHL are obtained prospectively. The specific requests for error bars in graphs and a Western blot for NFkB were met. It remains unclear whether patients with RpoC-

reactive NLPHL had persistent infection with *M. catarrhalis* (and thus persistent source of the MID/hag superantigen) or not.

*-> We thank reviewer 4 very much for the estimation of our manuscript. We agree that it would be interesting to know whether patients had persistent infection with *M. catarrhalis*, however, we consider this point to be beyond the scope of the current manuscript. Hopefully, in future we will be able to collect sterile cryospecimens of IgD⁺ NLPHL to analyze them for traces of *M. catarrhalis*. Furthermore, in such a future trial we would obtain nasal swabs and sputum of patients.*

Minor comments:

1. Text should refer to specific panels on figures rather than to the entire figure repeatedly.

-> We thank the reviewer for this helpful comment and now refer in the text to the respective subfigures.

2. It appears that the IgD+NLPHL which is associated with *M. catarrhalis* has specific clinicopathologic findings. However, in the second GHSG cohort, the authors find no association between high-titer RpoC reactivity and these findings. Is there an association between seropositivity and IgD+ status in that cohort? If these data are not available, this is a limitation of this study.

-> We agree with the reviewer that it would be interesting to correlate RpoC seropositivity and IgD expression in NLPHL. Unfortunately, the GHSG serum samples were derived from older cohorts, in which IgD status was not systematically recorded. This is a limitation of the study and we have mentioned this caveat in the Results section on page 7. We agree with the reviewer that it would be of high interest to collect prospectively serum samples together with the IgD-status in a future study.

3. Fig. 3: addition of a control stimulated by MID/hag alone would be persuasive, as RpoC and MID/hag are always present together in vivo, so the principal effect on the IgD+ NLPHL cells could be through MID/hag.

-> We thank the reviewer for this important advice. We have now included the results of an experiment with stimulation by different fragments of MID/hag with or without RpoC as Supplementary Figure 11 and into the Results section on page 8.

4. P9: I would not describe IgD+NLPHL as having “a high mutational burden” which usually

indicates a genomic mutation rate; the authors presumably refer to the rate of somatic hypermutation.

-> We agree with the reviewer and have changed the phrase in the Discussion section on page 9 accordingly to the proposal of the reviewer to “a high rate of somatic hypermutation”.

REVIEWERS' COMMENTS:

Reviewer #2 (Remarks to the Author):

The addition of the MID/HAG control to the DEV cell experiments helps in the interpretation of results. Given the inductions of DEV proliferation by both RpoC and MID/HAG together appear to be roughly the sum of those signals from them alone, it may help clarity if the authors were to provide stronger justification for the conclusion that the effects of the two are synergistic instead of additive .

Reviewer #4 (Remarks to the Author):

My previous comments have been addressed adequately.

REVIEWERS' COMMENTS:

Reviewer #2 (Remarks to the Author):

The addition of the MID/HAG control to the DEV cell experiments helps in the interpretation of results. Given the inductions of DEV proliferation by both RpoC and MID/HAG together appear to be roughly the sum of those signals from them alone, it may help clarity if the authors were to provide stronger justification for the conclusion that the effects of the two are synergistic instead of additive .

-> We agree with the reviewer and have modified the text of the manuscript accordingly from synergistic to additive stimulation (Abstract, Results page 8, Discussion page 11).

Reviewer #4 (Remarks to the Author):

My previous comments have been addressed adequately.

-> We thank the reviewer for his/her evaluation of our manuscript.